# Dual Roles of miR-10a-5p and miR-10b-5p as Tumor Suppressors and Oncogenes in Diverse Cancers

**DOI:** 10.3390/ijms26010415

**Published:** 2025-01-06

**Authors:** Rajan Singh, Se Eun Ha, Tae Yang Yu, Seungil Ro

**Affiliations:** 1Department of Physiology and Cell Biology, University of Nevada School of Medicine, Reno, NV 89557, USA; rajans@med.unr.edu (R.S.); seeunh@med.unr.edu (S.E.H.); taeyangy@unr.edu (T.Y.Y.); 2Division of Endocrinology and Metabolism, Department of Medicine, Wonkwang University School of Medicine, Iksan 54538, Republic of Korea; 3RosVivo Therapeutics, Applied Research Facility, 1664 N. Virginia St., Reno, NV 89557, USA

**Keywords:** microRNAs, miR-10a, miR-10b, cancer, tumor suppressor, oncogene, therapeutics

## Abstract

Cancer is a complex genetic disorder characterized by abnormalities in both coding and regulatory non-coding RNAs. microRNAs (miRNAs) are key regulatory non-coding RNAs that modulate cancer development, functioning as both tumor suppressors and oncogenes. miRNAs play critical roles in cancer progression, influencing key processes such as initiation, promotion, and metastasis. They exert their effects by targeting tumor suppressor genes, thereby facilitating cancer progression, while also inhibiting oncogenes to prevent further disease advancement. The miR-10 family, particularly miR-10a-5p and miR-10b-5p (miR-10a/b-5p), is notably involved in cancer progression. Intriguingly, their functions can differ across different cancers, sometimes promoting and at other times suppressing tumor growth depending on the cancer type and target genes. This review explores the dual roles of miR-10a/b-5p as tumor-suppressive miRNAs (TSmiRs) or oncogenic miRNAs (oncomiRs) in various cancers by examining their molecular and cellular mechanisms and their impact on the tumor microenvironment. Furthermore, we discuss the potential of miR-10a/b-5p as therapeutic targets, emphasizing miRNA-based strategies for cancer treatment. The insights discussed in this review aim to advance our understanding of miR-10a/b-5p’s roles in tumor biology and their application in developing innovative cancer therapies.

## 1. Introduction

Cancer is a complex genetic disorder characterized by dysregulated expression of both coding and non-coding RNA transcripts [1,2]. microRNAs (miRNA) are small non-coding RNAs, typically 20 to 25 nucleotides in length, that play pivotal roles in cellular processes by binding to specific coding mRNAs and regulating their translation into proteins. The discovery of miRNAs has revolutionized research, revealing their critical functions not only in organismal development but also in disease mechanisms, particularly cancer [3,4]. Emerging evidence highlights the dysregulation of miRNAs in cancer development and progression [5,6].

miRNAs exhibit dual functionality in cancer, acting as both tumor-suppressive miRNAs (TSmiRs) and oncogenic es miRNAs (oncomiRs) [5]. Their regulatory roles span key cellular processes such as proliferation, differentiation, and apoptosis, with defects in these pathways contributing to oncogenesis [7,8]. Recent studies have shed light on the miR-10 family (miR-10a-5p and miR-10b-5p collectively refer to miR-10a/b-5p), demonstrating their therapeutic potential in conditions like diabetes and gastrointestinal (GI) motility disorders where they restore insulin-secreting pancreatic β cells and GI pacemaker cells, the interstitial cells of Cajal [9,10,11,12,13].

In cancer, miR-10a/b-5p exhibit bifunctional roles, acting as TSmiRs in some contexts and as oncomiRs in others, depending on their target genes and the cellular environment [14,15,16,17,18,19]. For instance, miR-10a/b-5p in some cancers suppresses tumor growth and metastasis [20,21], whereas their overexpression has been implicated in enhancing the proliferation, migration, and invasion in malignancies in other cancers [17,22,23]. These dual roles are particularly evident in cancers such as gastric, colorectal, breast, and gynecological cancers, highlighting their complex regulatory functions in cancer biology [15,16,24,25]. Notably, the role of a miRNA, whether tumor-suppressive or oncogenic, can vary even within a single cancer type, highlighting their context-dependent functionality [16,24,25].

This functional dichotomy is closely linked to their targets, including Hox transcripts and Krüppel-like factor 11 (KLF11). Both Hox genes and KLF11 exhibit dual roles, acting as tumor suppressors or oncogenes depending on cellular context and oncogenic signals in various cancers [14,26,27]. The dual roles of miR-10a/b-5p in tumorigenesis underscore their complex and context-dependent regulatory mechanisms, highlighting their potential as both biomarkers and therapeutic targets in cancer.

This review provides a comprehensive overview of the dual roles of miR-10a/b-5p in cancer biology and their prospective applications in developing innovative cancer therapies.

## 2. Dual Roles of miR-10a/b-5p as Tumor Suppressors and Oncogenes

Figure 1 provides an overview of the dual roles of miR-10a/b-5p, functioning as TSmiRs or oncomiRs across various cancers and even within the same cancer type (Table 1 and Table 2).

## 3. miR-10a/b-5p as Tumor-Suppressive miRNAs (TSmiRs)

### 3.1. Chronic Myeloid Leukemia

miR-10a-5p functions as a TSmiR and regulates the expression of upstream stimulatory factor 2 (USF2) in chronic myeloid leukemia (CML) [29]. This study revealed reduced miR-10a-5p levels in CD34^+^ cells from CML patients. The downregulation of miR-10a-5p led to the overexpression of USF2, a transcription factor that promotes cell growth. Restoring miR-10a-5p levels in these cells decreased USF2 expression, reduced cell proliferation, and enhanced apoptosis, thereby confirming the tumor-suppressive role of miR-10a-5p in CML.

### 3.2. Esophageal Squamous Cell Carcinoma

miR-10a-5p functions as a TSmiR and inhibits cell proliferation and metastasis in esophageal squamous cell carcinoma (ESCC). Reduced levels of miR-10a-5p were observed in ESCC tissues and cell lines [30]. Overexpression of miR-10a-5p suppresses ESCC cell proliferation, migration, and invasion, both in vitro and in vivo. Mechanistically, miR-10a-5p directly targets TIAM1, an oncogene associated with tumor progression, leading to its downregulation. This interaction subsequently inhibits the Rac1 signaling pathway, which is crucial for cytoskeletal reorganization and cell motility. Restoring miR-10a-5p expression in ESCC cells inhibits tumor growth and metastasis [30].

### 3.3. Renal Cell Carcinoma

miR-10b-5p functions as a TSmiR, with its expression progressively reduced from normal kidney tissue to primary metastatic renal cell carcinoma (RCC) and further diminished in metastatic RCC [38]. This downregulation is associated with disease progression and poorer patient outcomes. Mechanistically, the loss of miR-10b-5p leads to the upregulation of cAMP-responsive element binding protein 1 (CREB1), an oncogene implicated in RCC pathogenesis [46]. In RCC, miR-10b-5p has been identified as a crucial regulator of cell invasion and metastasis by targeting HOXA3, a key effector within the FAK/YAP signaling pathway [70,71]. Notably, a recent study demonstrated that miR-10b-5p overexpression significantly reduced HOXA3 activity, reducing cell motility and invasive potential [70]. Furthermore, a review of miRNA signatures in RCC underscores the diverse roles of miR-10b-5p in modulating tumor progression, suggesting its potential as a biomarker and therapeutic target [71]. Collectively, these findings highlight the tumor-suppressive role of miR-10b-5p in RCC and suggest that restoring its expression could serve as a potential therapeutic strategy.

## 4. miR-10a/b-5p as Oncogenic miRNAs (oncomiRs)

### 4.1. Cholangiocarcinoma

miR-10a-5p functions as an oncomiR as it is upregulated and promotes tumor growth by activating the Akt signaling pathway in cholangiocarcinoma (CCA) [19]. Inhibition of miR-10a-5p in vitro and in vivo suppresses proliferation and induces apoptosis, thereby inhibiting CAA growth [19].

### 4.2. Granulosa Cell Tumors

miR-10a-5p functions as an oncomiR and promotes tumor development in granulosa cell tumors (GCT) by regulating the Akt and Wnt pathways. One study showed that miR-10a-5p is significantly upregulated in malignant GCT tissues [23]. Functional analyses revealed that miR-10a-5p enhances GCT progression by targeting the tumor suppressor PTEN, thereby activating the Akt and Wnt signaling pathways. In vivo experiments using GCT xenograft mice showed that miR-10a-5p overexpression accelerates tumor growth, while *mir-10a* knockout results in a less aggressive tumor phenotype, further supporting the oncogenic role of miR-10a-5p [23].

### 4.3. Acute Myeloid Leukemia

miR-10a-5p functions as an oncomiR, promoting cell proliferation in acute myeloid leukemia (AML) by downregulating the tumor suppressor gene PTEN. miR-10a-5p is upregulated in AML cells, leading to a decrease in PTEN expression [51]. This reduction in PTEN activates the PI3K/AKT signaling pathway, thereby enhancing AML cell proliferation. Additionally, miR-10a-5p has been identified as a potential therapeutic target and predictive biomarker for MDM2 inhibition in AML. Inhibiting miR-10a-5p may enhance the efficacy of MDM2 inhibitors, offering a promising strategy for treating specific leukemia subtypes [72].

### 4.4. Prostate Cancer

miR-10b-5p acts as an oncomiR by promoting cell proliferation and epithelial–mesenchymal transition (EMT) by regulating key signaling pathways in prostate cancer [64]. miR-10b-5p is upregulated by the long non-coding RNA CHRF in PC3 prostate cancer cells, which enhances aggressive tumor behavior [64]. Mechanistically, miR-10b-5p activates the GSK3β/AKT and NF-κB signaling pathways, both of which are critical for tumor cell survival, proliferation, and EMT. These findings highlight the oncogenic role of miR-10b-5p in prostate cancer by modulating pathways that drive tumor progression and metastasis.

### 4.5. Glioma and Glioblastoma

miR-10b-5p functions as an oncomiR and promotes proliferation, migration, invasion, and EMT in glioblastoma (GBM) [65]. TGF-β1, a critical regulator of GBM progression, upregulates miR-10b-5p expression, which targets key tumor suppressors like E-cadherin, Apaf-1, and PTEN, thereby enhancing tumor aggressiveness. Inhibition of miR-10b-5p with antagomir-10b-5p suppresses tumor growth in xenograft models [65]. Moreover, miR-10b-5p promotes glioma cell invasion by downregulating HOXD10 and upregulating invasion factors such as MMP14 and uPAR, highlighting the miR-10b-5p/HOXD10/MMP14/uPAR axis as a key driver of glioma malignancy [66]. Additionally, the long non-coding RNA GAS5 inhibits glioma progression by suppressing miR-10b-5p, indirectly affecting the Sirt1/PTEN/PI3K/AKT and MEK/ERK pathways [67]. GAS5-induced apoptosis and reduced motility are reversed by miR-10b-5p overexpression, further demonstrating the oncogenic role of miR-10b-5p in sustaining glioma growth and invasion [67]. These studies emphasize miR-10b-5p as a critical therapeutic target and its oncogenic role in GBM and glioma.

## 5. miR-10a/b-5p as Both TSmiRs and oncomiRs

### 5.1. Breast Cancer

miR-10a/b-5p play dual roles as TSmiRs and oncomiRs in breast cancer. As a tumor suppressor, miR-10a-5p inhibits tumor progression by targeting the PI3K/Akt/mTOR pathway, thereby reducing cell proliferation and invasion [25]. Similarly, miR-10b-5p suppresses breast cancer cell migration by downregulating TIAM1, a guanine nucleotide exchange factor for Rac1, leading to decreased Rac activation and reduced motility [35].

In contrast, miR-10b-5p also functions as an oncomiR in breast cancer. It promotes metastasis by targeting HOXD10 and upregulating RHOC, a gene critical for cell migration and invasion [57]. Elevated miR-10b-5p expression is associated with increased aggressiveness and metastatic potential in breast cancer tissues [54]. Clinical studies have demonstrated that elevated miR-10b-5p levels correlate with breast cancer metastasis, indicating its potential as a prognostic biomarker [73,74,75]. For instance, higher miR-10b-5p levels were observed in patients with metastatic breast cancer compared to those without metastasis [73]. Quantitative PCR analysis of 108 breast cancer tissue pairs demonstrated significantly elevated miR-10a/b-5p levels in tumors, with higher miR-10b-5p expression correlated with an increased risk of relapse [74]. Furthermore, innovative therapeutic strategies targeting miR-10b-5p, such as the nanodrug MN-anti–miR-10b, have demonstrated significant regression of metastatic breast cancer in preclinical models [75].

Additionally, miR-10b-5p enhances EMT and invasiveness by modulating TGF-β signaling [55,76]. Further evidence highlights miR-10b-5p’s role in driving proliferation and invasion by suppressing tumor suppressor genes and activating oncogenic pathways [56]. Collectively, these findings underscore miR-10b-5p as a pivotal oncogene in breast cancer, promoting EMT, invasion, and metastasis.

### 5.2. Bladder Cancer

miR-10b-5p exhibits dual roles as both a TSmiR and an oncomiR in bladder cancer. Its expression is significantly downregulated in bladder cancer tissues compared to adjacent normal urothelium, indicating a tumor-suppressive function [34]. This downregulation enhances tumor cell proliferation, migration, and invasion, whereas restoring miR-10b-5p levels in bladder cancer cells inhibits these oncogenic behaviors [34].

Conversely, miR-10b-5p also functions as an oncomiR in bladder cancer [69]. Elevated miR-10b-5p expression has been observed in bladder cancer cell lines and metastatic tissues, where it enhances cell migration and invasion [69]. Elevated miR-10a-5p levels are also associated with poor survival outcomes in bladder cancer. Furthermore, circulating plasma miR-10a-5p presents potential as a non-invasive diagnostic biomarker for bladder cancer patients [77]. Inhibition of miR-10b-5p has been shown to reduce these aggressive phenotypes [69]. Mechanistically, miR-10b-5p promotes metastasis by targeting tumor suppressor genes such as KLF4 and HOXD10, leading to the upregulation of invasion-related factors like MMP14 [69].

### 5.3. Endometrial Cancer

miR-10b-5p exhibits dual roles in endometrial serous adenocarcinoma, acting as both a TSmiR and an oncomiR. As a TSmiR, decreased miR-10b-5p expression is associated with vascular invasion, advanced tumor stage, and poor overall survival [39,40]. Its downregulation correlates with increased tumor aggressiveness and metastatic potential, likely by disrupting pathways that regulate cell proliferation and invasion. The loss of miR-10b-5p contributes to endometrial cancer development, underscoring its tumor-suppressive role in maintaining normal cellular regulation and its potential as a prognostic marker for disease severity and progression.

Conversely, miR-10b-5p also functions as an oncomiR by promoting tumor progression. It enhances cell proliferation, migration, and invasion and suppresses apoptosis by regulating Homeobox B3 (HOXB3) [68]. miR-10b-5p is upregulated in endometrial cancer tissues, coinciding with reduced HOXB3 expression [68]. Silencing the miR-10b-5p increases apoptosis and inhibits endometrial cancer cell proliferation, migration, and invasion. Mechanistically, miR-10b-5p targets HOXB3, as confirmed by dual-luciferase reporter assays. Overexpression of HOXB3 counteracted these oncogenic effects by promoting apoptosis and suppressing tumor growth and metastasis [68].

### 5.4. Cervical Cancer

miR-10a/b-5p exhibit dual roles as both TSmiRs and oncomiRs in cervical cancer. miR-10a/b-5p act as tumor suppressors in cervical cancer by targeting key oncogenes and signaling pathways. miR-10a-5p suppresses cancer cell proliferation by directly targeting brain-derived neurotrophic factor (BDNF), thereby inhibiting tumor progression [28]. Similarly, miR-10b-5p, often downregulated in cervical cancer tissues, inhibits cell proliferation, migration, and invasion by targeting insulin-like growth factor-1 receptor (IGF-1R) and Homeobox A1 (HOXA1) [21,36]. In HPV-positive cervical cancer, miR-10b-5p is downregulated through DNA methylation, and its overexpression suppresses cancer cell proliferation by targeting TIAM1 [37]. In small-cell cervical carcinoma, reduced miR-10b-5p expression correlates with advanced tumor stages, lymph node metastasis, and decreased survival [47]. Furthermore, miR-10b-5p downregulation during cervical cancer progression is associated with a more aggressive tumor phenotype [48]. These findings highlight the tumor-suppressive roles of miR-10a/b-5p in regulating oncogenic targets and pathways.

Conversely, miR-10a-5p can function as an oncomiR, promoting metastasis, angiogenesis, and tumor progression in cervical cancer. Elevated miR-10a-5p expression is associated with lymph node metastasis in primary tumor tissues [53]. miR-10a-5p mimics promote cancer cell migration and invasion by targeting the tumor suppressor PTEN, contributing to metastatic progression [53]. Furthermore, cancer-associated fibroblasts (CAFs) secrete extracellular vesicle (EV)-encapsulated miR-10a-5p, which is upregulated in cervical cancer tissues [52]. miR-10a-5p promotes tumor growth and angiogenesis by activating Hedgehog signaling through the downregulation of TBX5 [52]. Inhibition of miR-10a-5p in CAF-EVs effectively suppresses tumor growth and angiogenesis [52].

### 5.5. Ovarian Cancer

miR-10a/b-5p plays dual roles as TSmiRs and oncomiRs in ovarian cancer. As a TSmiR, miR-10a-5p targets oncogenes such as HOXA1 and GATA6, reducing cell proliferation, migration, and invasion, thereby inhibiting tumor progression [31,32].

Conversely, miR-10b-5p acts as an oncomiR. Its overexpression in ovarian cancer tissues and cell lines downregulates the tumor suppressor gene HOXD10, promoting enhanced migration and invasion of ovarian cancer cells driving tumor progression [50].

### 5.6. Gastric Cancer

miR-10b-5p exerts dual roles as both a TSmiR and an oncomiR in gastric cancer. As a tumor suppressor, miR-10b-5p targets the oncogenic proteins TIAM1 and MAPRE1 in gastric cancer cells [43,44,45]. Its downregulation in gastric cancer tissues leads to increased expression of TIAM1 and MAPRE1. Overexpression of miR-10b-5p inhibits cell proliferation, migration, and invasion while inducing apoptosis. Additionally, miR-10b-5p suppresses tumor growth in gastric cancer xenograft models by downregulating TIAM1 [42]. The CBFβ/RUNX3-miR-10b-5p-TIAM1 molecular axis further inhibits gastric cancer cell proliferation, migration, and invasion [43]. miR-10b-5p also epigenetically regulates MAPRE1, reinforcing its tumor-suppressive role in gastric cancer [45].

Conversely, miR-10b-5p acts as an oncomiR by promoting cell migration and invasion. Its upregulation in gastric cancer tissues and cell lines leads to the suppression of HOXD10, a tumor suppressor gene, resulting in increased expression of RHOC, a gene linked to enhanced metastatic potential [58]. Overexpression of miR-10b-5p facilitates the aggressive behavior of gastric cancer cells, highlighting its oncogenic role in tumor progression [58].

### 5.7. Colorectal Cancer

miR-10a/b-5p exhibit dual roles as TSmiRs and oncomiRs in colorectal cancer (CRC). As a TSmiR, miR-10a-5p regulates key processes involved in metastasis and epithelial integrity in CRC [24]. miR-10a-5p inhibits CRC metastasis by suppressing EMT and anoikis resistance by downregulating MMP14 and ACTG1 [24]. Additionally, the loss of miR-10a-5p activates lipocalin 2 and Wnt signaling, driving intestinal neoplasia in female mice, emphasizing its role in maintaining epithelial stability and preventing tumor progression [59]. Furthermore, miR-10b-5p suppresses the growth and metastasis of CRC by targeting FGF13 [41].

Conversely, miR-10b-5p acts as an oncomiR in CRC, promoting tumor progression and metastasis through multiple mechanisms [60,61,62]. Its upregulation in CRC tissues and cell lines enhances migration and invasion by targeting the tumor suppressor HOXD10, leading to increased RHOC expression [61]. Elevated miR-10b-5p levels are also associated with increased TWIST-1 expression and reduced E-cadherin levels, facilitating EMT and driving tumor progression [62]. Furthermore, miR-10b-5p promotes CRC growth and metastasis by downregulating P21 and P53 [60]. Clinically, high miR-10b-5p expression correlates with advanced-stage disease, liver metastasis, and aggressive tumor behavior in CRC patients [63].

### 5.8. Hepatocellular Carcinoma

miR-10a-5p exhibits dual roles as a TSmiR and an oncomiR in hepatocellular carcinoma (HCC). As a TSmiR, miR-10a-5p inhibits cell metastasis by targeting spindle and kinetochore-associated protein 1 (SKA1) in HCC tissues and cell lines [20]. Its expression is downregulated in HCC tissues and cell lines, and overexpression of miR-10a-5p suppresses HCC cell migration and invasion, both in vitro and in vivo [20]. Mechanistically, miR-10a-5p binds to the 3′ untranslated region of SKA1 mRNA, promoting its degradation and reducing SKA1 protein levels, thereby inhibiting the EMT process critical for cancer metastasis. Restoring miR-10a-5p expression in HCC cells effectively impedes tumor growth and metastasis [20].

Conversely, miR-10a-5p also functions as an oncomiR in HCC. It facilitates metastasis by downregulating PTEN and activating the PI3K/AKT/MMP2/MMP9 signaling axis [49]. Overexpression of miR-10a-5p in HCC tissues suppresses PTEN, leading to activation of the AKT pathway. This activation increases the levels of matrix metalloproteinases MMP2 and MMP9, which degrade the extracellular matrix and enhance tumor invasiveness [49].

While this review primarily focuses on miR-10a/b-5p, miR-34a-5p also plays significant roles in cancer regulation, functioning predominantly as a TSmiR [78]. Notably, miR-34a-5p can also exhibit oncogenic activity in certain cancers, such as papillary thyroid cancer and head and neck squamous cell carcinoma [79,80]. As a TSmiR, miR-34a-5p directly downregulates over 30 distinct oncogenes [78]. Experimental studies demonstrate that miR-34a-5p mimics reduce cancer cell proliferation, migration, and invasion in vitro, while inhibiting tumor growth, blocking metastasis, and improving survival in vivo [78]. MRX34, a liposomal formulation of the miR-34a-5p mimic, represents the first-in-class miRNA mimic cancer therapy [81] and has progressed to a Phase 1 clinical trial for patients with solid tumors [82]. Moreover, ten miRNA-based therapeutics are currently undergoing clinical evaluation in Phase 1 and Phase 2 trials [13]. These findings highlight the therapeutic potential of miR-10a/b-5p and underscore the broader promise of miRNA-based cancer treatments.

The paradoxical dual roles of miR-10a/b-5p in cancer highlight the complexity of miRNA-mediated regulation in tumor biology. These miRNAs demonstrate context-dependent functions, acting as TSmiRs by targeting oncogenes and inhibiting critical pathways such as EMT and metastasis while also functioning as oncomiRs by repressing tumor suppressor genes and activating oncogenic signaling pathways, including PI3K/Akt and Wnt. Their roles vary significantly across cancer types, including breast, bladder, endometrial, cervical, ovarian, gastric, colorectal, and hepatocellular cancers, depending on the molecular context and specific gene targets within the tumor microenvironment.

This dual functionality underscores the importance of precision medicine, which considers the unique tumor microenvironment and regulatory networks of miR-10a/b-5p in each cancer type. Future therapeutic strategies should focus on restoring their tumor-suppressive functions or inhibiting their oncogenic effects, paving the way for tailored and effective cancer treatments.

## 6. miR10a/b-5p in the Regulation of Key Cancer Pathways

miR-10a/b-5p play pivotal roles in regulating cancer-related signaling pathways, including PI3K/Akt/mTOR, Wnt/β-catenin, epithelial–mesenchymal transition (EMT), Hedgehog signaling, and intrinsic apoptotic pathways, by targeting either oncogenes or tumor suppressor genes. This summary highlights the key pathways influenced by miR-10a/b-5p, emphasizing their dual roles in cancer suppression and progression (Figure 2).

***PI3K/Akt/mTOR**:*** The PI3K/Akt/mTOR pathway is essential for cell growth, proliferation, and survival, with its dysregulation often associated with oncogenesis [83,84]. In glioblastoma, miR-10a-5p suppresses tumor growth by targeting components of the PI3K/Akt pathway, resulting in reduced cell proliferation and increased apoptosis [85]. In contrast, in HCC, miR-10a-5p acts as an oncomiR by downregulating PTEN, thereby activating the PI3K/Akt pathway and promoting metastasis [49].

***Wnt/β-catenin**:*** The Wnt/β-catenin pathway regulates cell fate determination, proliferation, and migration [86], with its aberrant activation linked to various cancers [87]. In colorectal cancer, miR-10a-5p suppresses metastasis by inhibiting EMT and anoikis resistance, primarily through downregulating MMP14 and ACTG1, which are key components of the Wnt/β-catenin pathway [24]. This pathway plays a crucial role in CRC progression, with miR-10a-5p serving as an important regulatory component [88]. Histone deacetylase inhibitors have been shown to attenuate Wnt signaling by depleting TCF7L2, a transcription factor essential for β-catenin-mediated gene transcription. miR-10a-5p further modulates this pathway by targeting upstream regulators of TCF7L2, thereby influencing the transcriptional output of Wnt signaling [88]. In glioma, miR-10a-5p promotes tumorigenesis by targeting myotubularin-related protein 3 (MTMR3) and modulating the Wnt/β-catenin signaling pathway, leading to increased cell proliferation and invasion [89].

***Epithelial–mesenchymal transition (EMT):*** EMT is a process by which epithelial cells acquire mesenchymal properties, enhancing their migratory capacity and invasiveness, thereby contributing to metastasis [90]. In gastric cancer, miR-10b-5p functions as a TSmiR by targeting oncogenic proteins such as TIAM1 and MAPRE1 [42,45]. Overexpression of miR-10b-5p inhibits proliferation, migration, and invasion while inducing apoptosis, thereby modulating the tumor microenvironment to suppress tumor progression. In breast cancer, however, miR-10b-5p promotes EMT by targeting HOXD10, increasing cell migration and invasion [57].

***Hedgehog signaling**:*** The Hedgehog pathway is essential for cancer as well as embryonic development and tissue regeneration [91]. In cervical cancer, miR-10a-5p promotes angiogenesis and tumor progression by activating Hedgehog signaling through the downregulation of TBX5 [52].

***TGF-β signaling:*** The TGF-β pathway regulates cell growth, differentiation, and apoptosis [92,93]. TGF-β suppresses tumor growth in early stages but promotes metastasis in advanced stages [93]. TGF-β1 upregulates miR-10b-5p expression, which downregulates the tumor suppressors E-cadherin, Apaf-1, and PTEN, thereby enhancing tumor aggressiveness in GBM [65]. In breast cancer, miR-10b-5p enhances invasiveness through modulation of TGF-β signaling, contributing to EMT and metastasis [76].

## 7. miR-10a/b-5p and Tumor Microenvironment

miR-10a/b-5p play a pivotal role in modulating the tumor microenvironment (TME) by regulating cellular functions within the tumor niche [94,95,96,97]. Extracellular miRNAs facilitate intercellular communication, interacting with stromal cells and extracellular matrix components to establish a microenvironment that supports tumor growth and immune evasion [98]. Among these, miR-10a/b-5p have emerged as critical regulators in various cancers. In HCC, miR-10a-5p inhibits cell metastasis by targeting SKA1 and suppressing the EMT process, a key driver of metastasis, thereby reducing the metastatic potential within the TME [20]. Conversely, in breast cancer, miR-10b-5p promotes metastasis by targeting HOXD10, which leads to RHOC upregulation—a gene crucial for cell migration and invasion—altering the TME to favor metastatic dissemination [57]. In gastric cancer, miR-10b-5p targets oncogenic proteins such as TIAM1 and MAPRE1, and its overexpression inhibits proliferation, migration, and invasion while inducing apoptosis, effectively modulating the TME to suppress tumor progression [42,44,45]. In CRC, miR-10b-5p is upregulated, enhancing cell migration and invasion by targeting the tumor suppressor HOXD10 and increasing RHOC expression, thereby supporting tumor aggressiveness [61].

These findings highlight the dual roles of miR-10a/b-5p in shaping the TME across different cancer types. By modulating key signaling pathways and gene expressions, these miRNAs influence tumor progression, metastasis, and the behavior of the tumor niche. Further research is needed to unravel their precise molecular mechanisms and functions within the TME. Such insights could pave the way for novel therapeutic strategies targeting miR-10a/b-5p to modulate the tumor microenvironment effectively.

## 8. miRNA Therapeutic Targets in Cancer

miR-10a/b-5p have emerged as pivotal regulators in cancer biology, demonstrating dual roles as TSmiRs and oncomiRs. This dual functionality positions them as promising therapeutic targets. The capacity of miRNAs to modulate multiple genes within the same pathway further underscores their therapeutic potential [99,100]. These multifaceted roles make them attractive candidates for novel cancer therapies.

Preclinical and in vitro studies have shown that reintroducing TSmiRs or inhibiting oncomiRs can effectively suppress cell migration and proliferation or induce apoptosis [13,101,102]. miRNA mimics can inhibit protein synthesis by binding to mRNA targets, while miRNA inhibitors can restore mRNA translation by binding to the miRNAs that repress it [13]. miRNA mimics are typically small double-stranded RNA and/or DNA molecules that associate with the RNA-induced silencing complex (RISC), guiding it to target mRNAs [13]. The mimic binds with imperfect complementarity to the target mRNA, blocking translation or causing mRNA degradation, leading to gene silencing [13]. In contrast, miRNA inhibitors are small single-stranded RNAs that bind to and suppress their target miRNAs, thereby restoring mRNA translation [13]. Thus, miRNAs, whether functioning as TSmiRs or oncomiRs could serve as therapeutic agents or targets. Unlike traditional gene-targeting therapies, modulating miRNA levels offers the advantage of simultaneously influencing multiple genes and pathways [103].

In cancers where miR-10a/b-5p act as oncomiRs, miRNA inhibitors can restore the synthesis of target tumor-suppressor proteins, thereby inhibiting tumor growth [104,105]. For instance, in metastatic breast cancer, miR-10b-5p inhibitors significantly reduced miR-10b-5p expression, leading to decreased cell migration and invasion, highlighting their potential for treating aggressive cancers [106]. Another study utilizing innovative therapeutic strategies targeting miR-10b-5p developed the nanodrug MN-anti–miR-10b, which demonstrated significant regression of metastatic breast cancer in preclinical models [75]. This approach highlights the potential of miR-10b-5p inhibitors in treating aggressive cancers.

miRNA sponges have emerged as another promising tool in miRNA therapeutics, leveraging their ability to sequester miRNAs and prevent their interactions with target mRNAs [105,107,108]. These synthetic constructs, comprising RNA molecules with multiple binding sites complementary to specific miRNAs, act as competitive inhibitors [107,108]. By binding and neutralizing oncomiRs, miRNA sponges effectively mitigate their downstream effects on tumorigenic pathways [107]. For example, advances in miRNA sponge technology have demonstrated their potential to selectively inhibit miR-21, a well-established oncomiR, significantly reducing tumor growth and progression in preclinical models [109]. Similarly, developing stable, cell-permeable miRNA sponges targeting miR-122 achieved notable attenuation of hepatocellular carcinoma progression [110]. In colorectal cancer, miRNA sponges targeting miR-21 and miR-155, two key oncomiRs, have shown remarkable efficacy in suppressing tumor growth and progression [111]. These findings highlight the potential of miRNA sponges, as innovative therapeutic tools for targeting oncomiRs, offering specificity and versatility in targeting oncogenic pathways across various cancers.

Conversely, in cancers where miR-10a/b-5p function as TSmiRs, miR-10a/b-5p mimics can be used to inhibit target oncogenes and suppress tumor progression. In CML, reduced miR-10a-5p levels led to the overexpression of oncogene USF2, thereby promoting tumor growth. Restoring miR-10a-5p levels by miR-10a-5p mimic in these cells decreased USF2 expression, reduced cell proliferation, and enhanced apoptosis, suggesting a viable therapeutic strategy for CML [29].

Advancements in miRNA delivery systems, including cancer-specific targeting strategies, along with a deeper understanding of their regulatory networks are crucial to unlocking their therapeutic potential. By modulating entire biological pathways, miRNA-based therapies represent a novel and highly promising class of cancer treatment, offering improved efficacy and specificity.

## 9. Conclusions and Future Perspectives

Cancer is a multifaceted genetic disorder involving alterations in both coding and non-coding RNA transcripts. Emerging evidence underscores the pivotal role of miR-10a/b-5p dysregulation in cancer development and progression. These miRNAs play dual roles in oncogenesis, functioning as either TSmiRs or oncomiRs depending on the cancer context and the specific genes or pathways they regulate.

The downregulation of miR-10a/b-5p is implicated in various cancers, including esophageal cancer, chronic myeloid leukemia, and renal cell carcinoma. In these contexts, miR-10a/b-5p mimics demonstrate therapeutic potential by targeting and inhibiting oncogenic factors such as TIAM1, USF2, and CREB1, thereby suppressing tumorigenicity (Figure 3). Conversely, the overexpression of miR-10a/b-5p is linked to cancers such as granulosa cell tumor, acute myeloid leukemia, and glioblastoma. In these cases, miR-10a/b-5p inhibitors may restore tumor suppressor pathways by modulating targets like PTEN, E-cadherin, HOXD10, Apaf-1, and SIRT1, thereby inhibiting tumor growth (Figure 3).

These findings highlight the potential of miR-10a/b-5p-based therapeutics, including mimics and inhibitors, for clinical applications. However, their successful translation into clinical use necessitates interdisciplinary efforts to enhance specificity, safety, and delivery mechanisms, particularly for cancer-specific targeting. Conflicting data regarding miR-10a/b-5p’s role in cancer, driven by variations in experimental design—such as differences in cancer models, sample sizes, and detection techniques—highlight the need for methodological standardization and innovative approaches for measuring miRNA levels. Particularly, both miR-10a-5p and miR-10b-5p levels should be evaluated in cancer as their conserved seed sequences suggest functional redundancy in targeting the same cancer pathways. Furthermore, the predominance of in vitro studies emphasizes the necessity of in vivo validation to elucidate their true functional roles within the complex landscape of cancer biology, paving the way for more robust early-stage clinical trial designs.

Further research is essential to clarify the dual roles of miR-10a/b-5p—tumor-suppressive or oncogenic—across specific cancer types. In cancers with conflicting evidence, their function may depend on the stage of cancer progression. Moreover, miR-10a/b-5p levels can act as either TSmiRs or oncomiRs in different cancers. While optimal levels of miR-10a/b-5p are essential for health and serve protective roles in preventing conditions such as diabetes, obesity, and gastrointestinal dysmotility [9], their overexpression has been linked to cancer progression [18]. Additionally, miR-10a/b-5p sponges may disrupt their roles as TSmiRs or oncomiRs. Unraveling these mechanisms is crucial for leveraging miR-10a/b-5p in precision oncology.

## Figures and Tables

**Figure 1 ijms-26-00415-f001:**
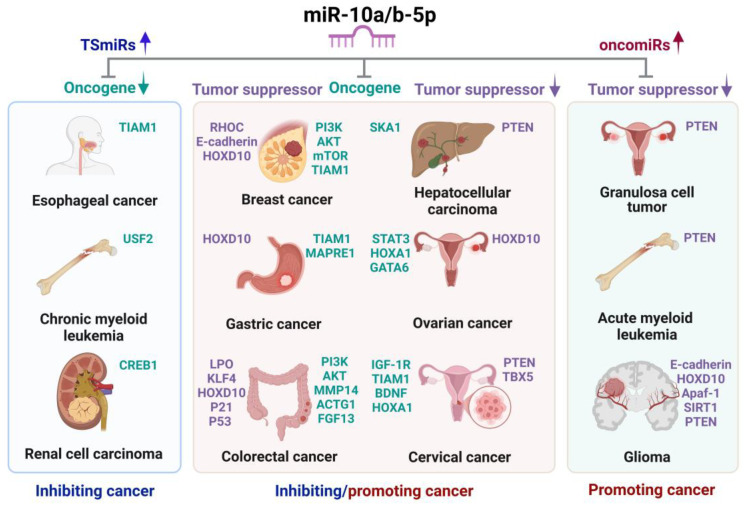
Dual roles of miR-10a/b-5p as tumor-suppressive miRNAs and/or oncogenic miRNAs in various cancers. miR-10a/b-5p function as either TSmiRs or oncomiRs depending on the cancer type and the molecular pathways they regulate. As TSmiRs, miR-10a/b-5p downregulate oncogenes, thereby inhibiting the progression of esophageal cancer, chronic myeloid leukemia, and renal cell carcinoma. Conversely, as oncomiRs, they downregulate tumor suppressor genes, promoting the development of granulosa cell tumor, acute myeloid leukemia, and glioma. Furthermore, miR-10a/b-5p exhibits dual functionality in breast, gastric, colorectal, hepatocellular, and ovarian cancers, acting as either TSmiRs or oncomiRs based on whether their target genes are oncogenes or tumor suppressors.

**Figure 2 ijms-26-00415-f002:**
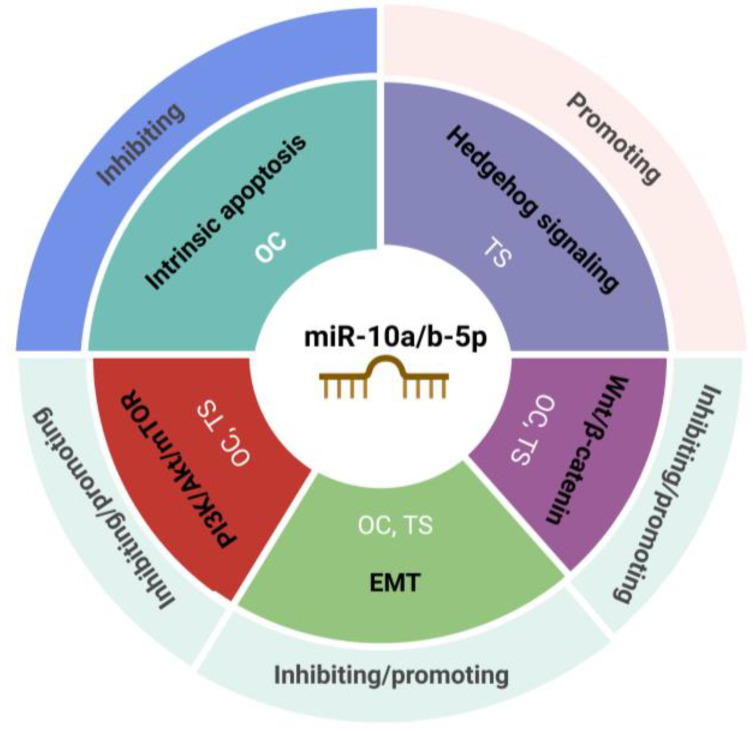
Cancer pathways regulated by miR-10a/b-5p. miR-10a/b-5p inhibit intrinsic apoptosis by targeting oncogenes. In contrast, they promote Hedgehog signaling by targeting tumor suppressor genes. Additionally, miR-10a/b-5p demonstrate dual functionality, either inhibiting or promoting the PI3K/Akt/mTOR axis, epithelial–mesenchymal transition (EMT), and Wnt/β-catenin signaling, depending on whether their targets are oncogenes (OC) or tumor suppressor genes (TS).

**Figure 3 ijms-26-00415-f003:**
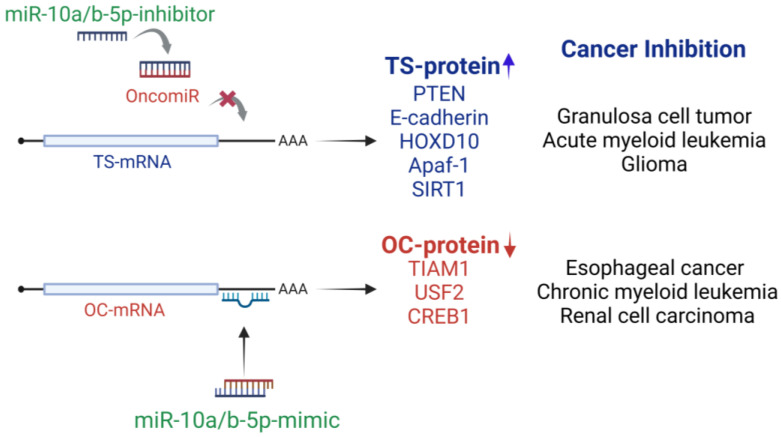
Therapeutic strategies targeting miR-10a/b-5p for cancer inhibition. miR-10a/b-5p inhibitors can counteract their oncogenic (oncomiR) functions, restoring tumor suppressor proteins and thereby suppressing bladder cancer, acute myeloid leukemia, and glioblastoma. Conversely, miR-10a/b-5p mimics can antagonize oncogenic mRNAs, reducing oncogenic protein expression and inhibiting esophageal cancer, chronic myeloid leukemia, and renal cell carcinoma. Abbreviations: oncomiR, oncogenic microRNA; TS, tumor suppressor; OC, oncogene.

**Table 1 ijms-26-00415-t001:** Roles and target genes of miR-10a/b-5p as tumor-suppressive miRNAs in human cancers and mouse models.

miRNA		Target Gene (Oncogene)	Cancer	Human			Mouse	Reference
Name	Role			Tissue (n)	Cell (n)	Blood (n)		
miR-10a-5p	TSmiR	PI3K/Akt/mTOR	Breast cancer (BC)		BC (2)			[25]
miR-10a-5p	TSmiR	AKT	Cholangiocarcinoma (CCA)		CCA (3)		CCA xenograft mice	[19]
miR-10a-5p	TSmiR	BDNF	Cervical cancer (CC)		CC (5)			[28]
miR-10a-5p	TSmiR	USF2	Chronic myeloid leukemia (CML)	CML (6) Bone marrow (85)	CML (5)			[29]
miR-10a-5p	TSmiR	MMP14/ACTG1	Colorectal cancer (CRC)	CRC (26)	CRC (2)			[24]
miR-10a-5p	TSmiR	TIAM1	Esophageal squamous cell carcinoma (ESCC)	ESCC (54)	ESCC (2)		ESCC xenograft mice Pulmonary metastasis mice	[30]
miR-10a-5p	TSmiR	SKA1	Hepatocellular carcinoma (HCC)	HCC (30)	HCC (4)	Plasma (32)		[20]
miR-10a-5p	TSmiR	HOXA1	Ovarian cancer (OC)	OC (56)	OC (4)			[31]
miR-10a-5p	TSmiR	GATA6	Ovarian cancer (OC)	OC (376)	OC (2)		OC xenograft mice	[32]
miR-10b-5p	TSmiR	STAT3	Ovarian cancer (OC)	OC (6)	OC (3)			[33]
miR-10b-5p	TSmiR		Bladder cancer (BLC)	BLC (77)				[34]
miR-10b-5p	TSmiR	TIAM1	Breast cancer (BC)		BC (4)			[35]
miR-10b-5p	TSmiR	HOXA1	Cervical cancer (CC)	CC (40)	CC (2)			[36]
miR-10b-5p	TSmiR	IGF-1R	Cervical cancer (CC)	CC (46)	CC (5)			[21]
miR-10b-5p	TSmiR	TIAM1	Cervical cancer (CC)	CC (70)	CC (3)			[37]
miR-10b-5p	TSmiR		Clear cell renal cell carcinoma (ccRCC)	ccRCC (250)				[38]
miR-10b-5p	TSmiR		Endometrial serous adenocarcinoma (ESA)	ESA (21)	ESA (1)			[39]
miR-10b-5p	TSmiR		Endometrioid endometrial carcinoma (EEC)	EEC (28)		Plasma (12)		[40]
miR-10b-5p	TSmiR	FGF13	Colorectal cancer (CRC)		CRC (3)		CRC xenograft mice	[41]
miR-10b-5p	TSmiR	TIAM1	Gastric cancer (GC)	GC (12)	GC (3)		GC xenograft mice	[42]
miR-10b-5p	TSmiR	TIAM1	Gastric cancer (GC)	GC (19)	GC (4)			[43]
miR-10b-5p	TSmiR	TIAM1	Gastric cancer (GC)	GC (100)	GC (4)			[44]
miR-10b-5p	TSmiR	MAPRE1	Gastric cancer (GC)	GC (32)	GC (11)			[45]
miR-10b-5p	TSmiR	CREB1	Renal cancer (RC)	RC (35)	RC (4)			[46]
miR-10b-5p	TSmiR		Small cell cervical carcinoma (SCCC)	SCCC (44)				[47]
miR-10b-5p	TSmiR		Cervical cancer (CC)	CC (44)				[48]

TSmiR, tumor-suppressive miRNA; n, number; KO, knockout.

**Table 2 ijms-26-00415-t002:** Roles and target genes of miR-10a/b-5p as oncogenic miRNAs in human cancers and mouse models.

miRNA		Target Gene (Tumor Suppressor)	Cancer	Human		Mouse	Reference
Name	Role			Tissue (n)	Cell (n)		
miR-10a-5p	oncomiR	PTEN	Granulosa cell tumor (GCT)		Granulosa cells (2)	*mir-10a* KO mice GCT xenograft mice	[23]
miR-10a-5p	oncomiR	PTEN	Hepatocellular carcinoma (HCC)	HCC (30)	HCC (1)		[49]
miR-10b-5p	oncomiR	HOXD10	Ovarian cancer (OC)	OC (68)	OC (3)		[50]
miR-10a-5p	oncomiR	PTEN	Acute myeloid leukemia (AML)	AML (60)	AML (1)		[51]
miR-10a-5p	oncomiR	TBX5	Cervical squamous cell carcinoma (CSCC)	CSCC (60)	CSCC (2)	CSCC xenograft mice	[52]
miR-10a-5p	oncomiR	PTEN	Cervical cancer (CC)	CC (40)	CC (2)		[53]
miR-10b-5p	oncomiR	E-cadherin	Breast cancer (BC)	BC (45)	BC (2)	BC xenograft mice	[54]
miR-10b-5p	oncomiR	IQGAP2	Triple-negative breast cancer (TNBC)	TNBC (42)	TNBC (3)		[55]
miR-10b-5p	oncomiR	E-cadherin	Breast cancer (BC)	BC (44)	BC (1)		[56]
miR-10b-5p	oncomiR	HOXD10	Breast cancer (BC)	BC (18)	BC (6)		[57]
miR-10b-5p	oncomiR	HOXD10	Gastric cancer (GC)	GC (436)	GC (7)		[58]
miR-10a-5p	oncomiR	LPO/KLF4	Colorectal cancer (CRC)	CRC (16)		CRC (Apc) mice *mir-10a* KO mice	[59]
miR-10b-5p	oncomiR	P21 and P53	Colorectal cancer (CRC)	CRC (63)	CRC (5)	CRC xenograft mice	[60]
miR-10b-5p	oncomiR	HOXD10	Colorectal cancer (CRC)	CRC (70)			[61]
miR-10b-5p	oncomiR	E-cadherin	Colorectal cancer (CRC)	CRC (50)	CRC (1)		[62]
miR-10b-5p	oncomiR		Colorectal cancer (CRC)	CRC (246)			[63]
miR-10b-5p	oncomiR	GSK3β	Prostate cancer (PC)		PC (2)		[64]
miR-10b-5p	oncomiR	Apaf-1, E-cadherin	GBM	GBM (15)	GBM (2)		[65]
miR-10b-5p	oncomiR	HOXD10	Glioma	Glioma (22)	Glioma (4)		[66]
miR-10b-5p	oncomiR	SIRT1	Glioma		Glioma (2)		[67]
miR-10b-5p	oncomiR	HOXB3	Endometrial cancer (EC)	EC (20)			[68]
miR-10b-5p	oncomiR	KLF4/HOXD10	Bladder cancer (BLC)	BLC (20)	BLC cancer (6)		[69]

oncomiR, oncogenic miRNA; n, number; KO, knockout.

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
