# Peer review of "Dual Roles of miR-10a-5p and miR-10b-5p as Tumor Suppressors and Oncogenes in Diverse Cancers"

_ijms, 2025, doi:10.3390/ijms26010415_

Round 1

Reviewer 1 Report

Comments and Suggestions for Authors

The review article titled “Dual roles of miR-10a and miR-10b as tumor suppressors and oncogenes in diverse cancers” presents an overview of the dual roles of microRNAs miR-10a and miR-10b (miR-10a/b) in cancer biology. The authors discuss how these miRNAs can function as both tumor suppressors (TSmiRs) and oncogenes (oncomiRs) in various cancer types, depending on the cellular context and molecular environment. The review explores the mechanisms by which miR-10a/b regulates key cancer pathways, their impact on the tumor microenvironment, and their potential as therapeutic targets in cancer treatment. The article begins by introducing the importance of miRNAs in cancer development and progression. It then delves into the specific roles of miR-10a/b across different cancer types, including chronic myeloid leukemia, esophageal squamous cell carcinoma, renal cell carcinoma, cholangiocarcinoma, and various others. The authors provide detailed explanations of how miR-10a/b can act as TSmiRs or oncomiRs in each type of cancer, often within the same cancer. The review also discusses the involvement of miR-10a/b in regulating critical cancer pathways such as PI3K/Akt/mTOR, Wnt/β-Catenin, epithelial-mesenchymal transition (EMT), Hedgehog signaling, and TGF-β signaling. Furthermore, it explores the role of these miRNAs in modulating the tumor microenvironment and their potential as therapeutic targets in cancer treatment. The review is organized logically, with clear sections addressing different aspects of miR-10a/b in cancer biology. The reviewer has the following comments to the authors that need to be addressed.

1.     The review would benefit from incorporating more quantitative data to substantiate the qualitative descriptions of miR-10a/b's effects. Additionally, a more critical analysis of conflicting results or limitations in the current research on miR-10a/b would strengthen the manuscript and provide a balanced perspective on the topic.

2.     The figures in the review article are primarily general and illustrative in nature. Incorporating figures that present quantitative data would enhance the article’s depth and provide readers with valuable, data-driven insights

3.     The article predominantly focuses on miR-10a/b without providing significant comparisons to other cancer-related miRNAs, which could offer valuable context. For instance, miR-34a is well-documented for its dual roles as both a tumor suppressor and an oncogene. Recent advancements in the targeted delivery of miR-34a to suppress oncogenes highlight its therapeutic potential. To enrich the discussion and enhance the comprehensiveness of the manuscript, the authors are encouraged to incorporate references to the following relevant studies.

https://www.nature.com/articles/s41388-023-02801-8

https://www.cell.com/molecular-therapy-family/nucleic-acids/fulltext/S2162-2531(24)00080-5

4.     The article provides a thorough overview of preclinical research; however, it lacks sufficient discussion on clinical studies involving miR-10a/b. To provide a more comprehensive perspective, the authors are encouraged to include information on any available clinical studies related to miR-10a/b. This addition would significantly enhance the manuscript's relevance and applicability to clinical contexts.

5.     The manuscript references tables and their legends within the text; however, the actual tables are missing. The authors are advised to include the referenced tables.

Author Response

Response letter

We sincerely appreciate the thoughtful feedback and valuable suggestions provided by the reviewers to improve our manuscript. Below, we provide detailed point-by-point responses to the reviewers' comments. The revised manuscript has been submitted as both a clean version and a track change version for reference. 

Point-by-point response

Reviewer 1

The review article titled “Dual roles of miR-10a and miR-10b as tumor suppressors and oncogenes in diverse cancers” presents an overview of the dual roles of microRNAs miR-10a and miR-10b (miR-10a/b) in cancer biology. The authors discuss how these miRNAs can function as both tumor suppressors (TSmiRs) and oncogenes (oncomiRs) in various cancer types, depending on the cellular context and molecular environment. The review explores the mechanisms by which miR-10a/b regulates key cancer pathways, their impact on the tumor microenvironment, and their potential as therapeutic targets in cancer treatment. The article begins by introducing the importance of miRNAs in cancer development and progression. It then delves into the specific roles of miR-10a/b across different cancer types, including chronic myeloid leukemia, esophageal squamous cell carcinoma, renal cell carcinoma, cholangiocarcinoma, and various others. The authors provide detailed explanations of how miR-10a/b can act as TSmiRs or oncomiRs in each type of cancer, often within the same cancer. The review also discusses the involvement of miR-10a/b in regulating critical cancer pathways such as PI3K/Akt/mTOR, Wnt/β-Catenin, epithelial-mesenchymal transition (EMT), Hedgehog signaling, and TGF-β signaling. Furthermore, it explores the role of these miRNAs in modulating the tumor microenvironment and their potential as therapeutic targets in cancer treatment. The review is organized logically, with clear sections addressing different aspects of miR-10a/b in cancer biology. The reviewer has the following comments to the authors that need to be addressed.

The review would benefit from incorporating more quantitative data to substantiate the qualitative descriptions of miR-10a/b's effects. Additionally, a more critical analysis of conflicting results or limitations in the current research on miR-10a/b would strengthen the manuscript and provide a balanced perspective on the topic.

Response: Thanks for these thoughtful comments. This revision contains Tables 1 and 2, which provide miR-10a-5p or miR-10b-5p, its role, target genes, cancer types, sample types and sizes (numbers) in humans and mice as quantitative data. These tables were originally submitted with the manuscript, but somehow not delivered to the reviewers. Furthermore, we added a discussion addressing conflicting data, study limitations, and future perspectives.  

The figures in the review article are primarily general and illustrative in nature. Incorporating figures that present quantitative data would enhance the article’s depth and provide readers with valuable, data-driven insights

Response: We updated the figures by indicating the up- or down-regulation of oncogenes and tumor suppressors by TSmiRs and oncomiRs in Figure 1 as well as miR-10a/b-5p mimics and miR-10a/b-5p inhibitors in Figure 3, showing antagonistic interactions between miRNAs and target genes.

The article predominantly focuses on miR-10a/b without providing significant comparisons to other cancer-related miRNAs, which could offer valuable context. For instance, miR-34a is well-documented for its dual roles as both a tumor suppressor and an oncogene. Recent advancements in the targeted delivery of miR-34a to suppress oncogenes highlight its therapeutic potential. To enrich the discussion and enhance the comprehensiveness of the manuscript, the authors are encouraged to incorporate references to the following relevant studies.

https://www.nature.com/articles/s41388-023-02801-8

https://www.cell.com/molecular-therapy-family/nucleic-acids/fulltext/S2162-2531(24)00080-5

Response: Thanks for this suggestion. We included a discussion on the roles of miR-34a in cancer and its clinical advancements as a therapeutic target lung and liver cancer.

The article provides a thorough overview of preclinical research; however, it lacks sufficient discussion on clinical studies involving miR-10a/b. To provide a more comprehensive perspective, the authors are encouraged to include information on any available clinical studies related to miR-10a/b. This addition would significantly enhance the manuscript's relevance and applicability to clinical contexts.

Response: Thanks for the suggestion. In the revised manuscript, we included the clinical studies utilizing therapeutic strategies targeting miR-10a/b-5p in breast and bladder cancer.

The manuscript references tables and their legends within the text; however, the actual tables are missing. The authors are advised to include the referenced tables.

Response: This revision includes the two tables. They were originally submitted with the manuscript, but somehow not delivered to the reviewers.

Reviewer 2

Recommendation for Manuscript Revision and Publication

I recommend this manuscript for publication in the journal, provided the authors address the following major revisions to enhance its clarity, scientific rigor, and overall impact:

  1. Graphical Representation

To improve the manuscript’s impact and comprehensibility, the authors should provide a graphical representation illustrating the role of miR-10a/b in different types of cancers, along with detailed mechanisms. A cartoon-style schematic diagram showcasing the pathways and molecular interactions would enhance the review’s appeal and facilitate better understanding by readers.

Response: We thank for this insightful feedback. In response to the recommendation, we updated Figures 1, 2, and 3 to show the dual roles of miR-10a/b-5p, cancer pathways, and therapeutic strategies. Figure 1 can be used as graphical representation.

  1. Inclusion of Recent Publications

The manuscript currently lacks references to several recent and relevant studies. The authors should update the reference list and incorporate findings from these studies into the discussion to provide a comprehensive overview. For example:

miR-10b suppresses cell invasion and metastasis through targeting HOXA3 regulated by FAK/YAP signaling pathway in clear-cell renal cell carcinoma.

BMC Nephrology, Volume 20, Article number: 127 (2019). Additionally, the manuscript should incorporate existing review articles on this topic to strengthen its foundation. Suggested inclusion: MicroRNA Signature in Renal Cell Carcinoma.

Front Oncol. 2020 Nov 30;10:596359. doi: 10.3389/fonc.2020.596359.

Response: Thanks for these suggestions. We included the findings from the recommended studies in the revised manuscript to provide a more comprehensive and updated discussion.

  1. Regulation of Key Cancer Pathways

In the section “miR-10a/b in the regulation of key cancer pathways”, the paragraph discussing the Wnt/β-Catenin pathway should include additional details on the impact of miR-10a on the TCF7L2 gene. Specifically, the authors should reference:

Histone deacetylase inhibitors induce attenuation of Wnt signaling and TCF7L2 depletion in colorectal carcinoma cells.

International Journal of Oncology. doi: 10.3892/ijo.2014.2550.

Incorporating this information will provide a more thorough explanation of the interaction between miR-10a and wnt signalling.

Response: We incorporated the key findings from the recommended study to enhance our discussion on the Wnt/β-Catenin pathway and its regulation by miR-10a-5p.

  1. Therapeutic Applications of miRNAs in Cancer

In the section “miRNA therapeutic targets in cancer”, the authors should expand their discussion on the use of miRNAs as tools in cancer treatment. The following aspects should be elaborated:

miRNA Inhibition:

Methods such as AntagomiRs, which are oligonucleotides designed to bind specific miRNAs.

Stable RNA molecules with multiple binding sites for target miRNAs, effectively sequestering them from their mRNA targets. Lots of publications are available.

miRNA Replacement Therapy:

The role of miRNA mimics in restoring tumor-suppressor miRNA functions.

The authors should also provide more detailed mechanisms for these therapeutic approaches. A revised graphical representation should move beyond a simple depiction of "promotion" or "inhibition" and include pathway-level mechanisms in a visually intuitive manner.

Addressing these revision will significantly enhance the manuscript clarity and scientific quality.

Response: We thank the thoughtful suggestion. In the revised manuscript, we included miRNA sponges and therapeutic approaches for cancer. Additionally, we included the targeting mechanisms and therapeutic strategies of miRNA mimics and inhibitors.

Additionally, we revised Tables 1 and 2 with citations incorporated in the reference list, as well as editing or correcting typos throughout this manuscript.

Reviewer 2 Report

Comments and Suggestions for Authors

This review article is well-written. Most of the topics on miR-10a/b are covered in this article. However, modification of few paragraphs and figures could enhance the scientific quality of the article. 

Author Response

(The authors gave the same response as above.)
